# Revelation in the Muslim, Christian, and Jewish Traditions: A Critical Analysis of Muhammad Abduh's Notion of Revelation through the Lens of Joseph Ratzinger and Benedict De Spinoza

**Mohammad Syifa Amin Widigdo** [1,*] and **Abd Razak Bin Zakaria** [2]

1   Program Studi Ekonomi Syariah, Universitas Muhammadiyah Yogyakarta, Yogyakarta 55000, Indonesia
2   Department of Educational Foundations and Humanitie, University of Malaya,
    Kuala Lumpur 50630, Malaysia; abdrazak@um.edu.my
*   Correspondence: syifamin@umy.ac.id

**Abstract:** This research investigated one of the foundational notions of religion, i.e., revelation, as presented by Muhammad Abduh (d. 1905) in *Risālat al-Tawḥīd* and *Risālat al-Wāridāt*, through a comparison with the understanding of revelation in the Catholic tradition, as elucidated by Pope Ratzinger (b. 1927), and in Judaism, as presented by the Jewish scholar B. D. Spinoza (d. 1677). This research closely considered Abduh's works to reveal whether the notion of revelation in the Islamic tradition is different from or analogous to its counterparts in Catholicism and Judaism. Although these authors' religious backgrounds are diverse, their understandings of revelation are analogous in the sense that revelation is understood as beyond the linguistic realm. However, they each have different religious and intellectual stances regarding the valid interpretation and knowledge of revelation; where Ratzinger relies on the Church's authority, Spinoza believes in the efficacy of holy scripture, and Abduh has more confidence in the use of reason in understanding revelation. By delineating the commonalities and differences of the ideas of these three scholars from different religious backgrounds, a more open and fruitful interreligious conversation can be further cultivated.

**Keywords:** revelation; Abduh; Ratzinger; Spinoza; Islam; Catholicism; Judaism

Kung (2007), a Roman Catholic priest and Professor at Tubingen University in Germany, once said, "no peace among the nations without peace among the religions. No peace among the religions without dialogue between the religions. No dialogue between the religions without investigating the foundations of the religions".[1] One of the challenges for world peace is that violence is often committed in the name of a religion, by citing its scriptural sources. Islam employs Qur'anic justifications, while Christianity and Judaism cite biblical verses. In this regard, as Kung suggests, the investigation of the foundational teachings of religions is important for understanding their peculiarities and commonalities and for building a solid foundation for peaceful civilization.[2] Following this suggestion, this study discusses one of the foundational elements in religion, i.e., revelation. The concept of revelation relies on the context of an engagement between that which is considered eternal and temporal reality.

For the majority of Muslims, especially after the fall of Muʿtazilī's doctrines and the triumph of Ashʿarī's theology, revelation (*waḥy*) is considered to be the eternal words of God.[3] God, as the eternal being, discloses the knowledge of religious teachings to His prophet; therefore, the knowledge communicated through (and embodied in) such words is also deemed to be eternal. Although the Islamic modernist Muhammad Abduh (d. 1905)[4] tried to reform such an understanding, he also intended to establish an adequate justification for the necessity of revelation.

However, Abduh's notion of revelation in *Risālat al-Tawḥīd* is less studied than his call for social and theological reform. Shabir and Susilo (2018) examine traces of Abduh's educational reform in Indonesia.[5] In a similar vein, Kevin W. Fogg (2015) provides an

annotated translation of an honorary doctoral speech of an Indonesian scholar, Haji Abdul Malik Karim Amrullah (Hamka), that highlights the influence of Muhammad Abduh on Islamic thought in Indonesia.[6] In another study, Oliver Scharbrodt (2007) discusses the mystical aspects of Muhammad Abduh in *Risālat al-Wāridāt*.[7] In Omer Aydin (2005), he argues that Abduh's inclination towards free will is stronger than fatalism.[8] John W. Livingston (1995) emphasizes that Abduh's favoring of reason plays a reconciliatory role in the relationship between science and religion.[9] Yusuf H. R. Seferta (1985) addresses the topic of prophethood in Abduh's *Risālat al-Tawḥīd* by comparison with Rashid Rida's prophecy in *al-Manār*.[10] These studies mainly elaborate social, educational, theological, or mystical aspects of Abduh's thought and influence. The study that most closely deals with the notion of revelation is Seferta's discussion of Abduh's justification of the necessity of prophethood (*nubuwwah*) and that which accompanies such attainment: revelation. While Seferta's study provides a good introduction on how Abduh perceives revelation as it relates to the notion of prophecy, it leaves a lacuna, since the concept of revelation in Abduh's work is inextricably linked with his discussion of reason and knowledge.

Therefore, this study aims to better understand the notion of revelation as it relates to knowledge, reason, faith, and imagination, through a comparative approach. I argue that a better understanding of revelation in Abduh's *Risālat al-Tawḥīd* could be reached by comparison with other traditions. To that end, Abduh will be compared with Joseph Ratzinger (1927–present), representing Catholic theologians, and Benedict De Spinoza (d. 1677), as a Jewish philosopher. Joseph Ratzinger, in 2006, delivered a speech at the University of Regensburg, Germany, in the context of discussing faith and reason.[11] By citing Theodore Khoury's work on the dialogue between the Byzantine Emperor, Manuel II Palaeologus, and an anonymous Persian scholar, he set a contrast between Catholic and Muslim attitudes toward reason and rationality in religious faith. While Christianity, for him, was inseparably linked with logos, reason, and rational arguments, Islam was described as inimical to reason and rationality.[12] Although one might question the choice of Ratzinger in this comparative study over John Paul II's encyclical *Fides et Ratio* (Paul 1998)[13], in the Regensburg speech, Ratzinger addressed Islam, and his treatment of revelation therein had both commonalities and differences with Abduh's thought, whereas one primarily finds commonalities between John Paul II's *Fides et Ratio* and Abduh's work on the discussion of reason. As for Spinoza, he not only inherited Greek philosophical thought through his Jewish (Maimonides, d. 1204) and Muslim (Abū Naṣr al-Fārābī, d. 950) teachers,[14] but his thought also influenced Muslim scholars in the Muslim world, especially in Iran.[15] From the intellectual encounters of Ratzinger and Spinoza with the Muslim world and scholarship, it is interesting to discuss their thoughts of revelation as compared to Abduh's ideas; in this way, an honest and fruitful inter-faith conversation can be further fostered and pursued.

From this perspective, their discussions of revelation will be viewed through a phenomenological lens, which, first, presents Abduh's, Ratzinger's, and Spinoza's concepts of revelation "on the bracket"(meaning refraining any judgments (*epoche*)); second, this approach presents these concepts in a fresh way and transcends the differences between them through eidetic intuition; and third, it provides a new understanding of revelation, through emphatic intersubjectivity.[16] As a result, revelation becomes inextricably associated with knowledge whose validity may be obtained or justified through reason (Abduh), faith (Ratzinger), and imagination (Spinoza).

## 1. What Is Revelation?

### a. Definition

To begin with, Muhammad Abduh uses the term "*waḥy*" to refer to the reality of revealing something to others, while Ratzinger and Spinoza use the word "revelation" which originally comes from the Latin word, '*revelation*', to designate such reality. Revelation

in Latin word means "the act of removing the veil, of unveiling something, or showing something".[17]

Pope Benedict XXVI employs the word revelation precisely in the light of reality beyond the material realm. He states "Revelation means God's whole speech and action ... A scripture is the material principle of revelation ... , but that it is not revelation itself".[18] Therefore, revelation in Ratzinger's elucidation is more than scripture. It is the *pneuma* (spirit) rather than the *gramma* (letter).

Spinoza is more straightforward in referring to revelation as an immaterial entity that he calls knowledge. For Spinoza, revelation is not common knowledge. Rather, it is "certain knowledge of something, revealed by God to men".[19] In another phrase, he also described revelation as the things revealed by God that extend beyond the limits of natural knowledge.[20] He differentiated natural knowledge from prophetic knowledge. The first is common to all men, but the latter specifically belonged to the Prophet who should interpret the things revealed by God to those who cannot have specific knowledge of them.[21]

Similarly, in the *Risālat al-Tawḥīd*, Muhammad Abduh perceives revelation (*waḥy*) as inner-knowledge (*ʿirfān*), not as words, speech, or action. Abduh defined *waḥy* as "inner-knowledge (*ʿirfān*) found by someone within himself with the assurance that it has come from God with mediation or without mediation".[22] Here Abduh chose the word "*ʿirfān*" instead of *iʿlām* (disclosing information or knowledge) or *kalām* (speech), which Muslim theologians usually use. Revelation, then, is perceived in its cognitive and spiritual aspects rather than in its material forms.[23] In this regard, Abduh seemed to be influenced by a gnostic (*ʿirfān*) intellectual tradition in the Islamic world. In the eastern land of Islam, the term *ʿirfan*, which is usually translated as gnosis, refers to what Alexander Knysh called "synthesis of philosophy, speculative theology, and mystical thought that emerged in the later medieval period and has persisted until today".[24] In the case of Abduh, the mystical-philosophical dimension of *irfan* (gnosis) is more apparent than its theological (*kalām*) dimension, which he inherited from his intellectual mentor, Jamaluddīn al-Afghānī (d. 1897).[25] Albert Hourani recorded that Abduh studied mystical commentaries of the Qurʾan and a philosophical work, *Ishārāt wa tanbīhāt* of the influential Muslim philosopher Ibn Sīnā (d. 1037), under the tutelage of al-Afghani.[26]

### b.  Reception and Justification of Revelation

Regarding how prophets receive revelation and how it can be justified, the Pope explained that because revelation in Christianity is Christ himself, the reception of revelation is equivalent to entering into the Christ-reality, either by "Christ in us" or "we in Christ".[27] In other words, it could be said that revelation is acquired through "entering", incarnation, or union between a prophet and God. Thus, in terms of justifying the phenomena of revelation, Ratzinger would use faith instead of reason because revelation "always and only becomes reality where there is faith".[28] Therefore, the justification of revelation in which the Christ-reality becomes ours is called "faith" in biblical language.[29]

Meanwhile, Spinoza believed that prophets receive revelation not through perfection of mind, but instead through the power of imagination. He wrote, "The Prophets perceived God's revelation only with the aid of the imagination, that is, by the meditation of words or images, the latter of which might be true or imaginary".[30] According to Spinoza, this issue might be because prophets were endowed, not with a more perfect mind, but instead, with a power of imagining unusually vividly.[31] The certainty and validity of imagination, according to Spinoza, can be guaranteed through a sign from God. For example, in the case of Abraham, he asked for a sign showing that it was God who had made this promise to him.[32]

In terms of how people can accept the possibility of revelation, Spinoza tended to agree with Ratzinger. While prophets were endowed with "a power of imagination" in acquiring revelation, Spinoza maintained that people could accept such phenomena not through imagination. Instead, they can justify the reality of revelation through what Spinoza called

"a sheer faith"[33], especially the faith of the truth of what is revealed by God to the prophets. According to this philosopher, believing in the truth of revelation is important because knowledge of all the things in Scripture must be only sought from Scripture itself, just as the knowledge of nature must be sought from nature itself.[34] In short, for Spinoza, "faith" and "Scripture" are significantly important for the understanding of the phenomena revealed by God.

Meanwhile, Muhammad Abduh in *Risālat al-Tawḥīd (Theology of Unity)* acknowledged that among human beings, there are men who have a higher purity of the soul (*naqā' al-jawhar*) and inward disposition (*aṣl al-fiṭrah*). According to Abduh, such noble men can see the things of God as if by natural vision, while others could reach this neither by reason nor sense, nor even with the aid of proof and demonstration.[35] Ultimately, they can only attain the highest level of human life by virtue of Divine emanation (*al-fayḍ al-ilāhī*). Those men are prophets. Abduh maintained that nothing could forbid the souls of the prophets (*nufūs al-anbiyā'*) from perceiving the Divine knowledge given their noble quality.[36] In other words, Abduh seems to believe that the high quality of souls (*an-nufūs al-ʿāliyah*)—which he uses interchangeably with a high quality of intellect (*al-ʿuqūl al-sāmiyah*)—can receive the Divine knowledge.

As for human beings, in general, they are assumed to be able to understand the phenomena of revelation through intellectual reasoning. By this kind of reasoning, Abduh insisted that people can acquire a sound belief.[37] For him, grasping the phenomena of revelation makes people aware of the significance and the secret of reason. However, people who "do not want to understand and then suppress their own intelligence not to understand" will face difficulties apprehending it.[38] He criticized those who are doubtful about revelation as degrading humanity because of their mere reliance on the senses. Therefore, to Abduh, it is reason that can justify the possibility of revelation for the prophets.

## 2. Understanding of Human Knowledge, Reason, and Imagination

To translate "*waḥy*" into "revelation" in this light is probably acceptable since both are analogous in meaning. Abduh defines the term "*waḥy*" according to its etymological root.[39] The word "*waḥy*" is a verbal noun (*maṣdar*) of Arabic verb *waḥā*. By saying "*waḥayta ilayhi*", it means one revealed something to someone. In other words, *waḥy* means one spoke to somebody about something kept hidden from others.[40] The term "revelation" interestingly refers to a parallel reality. The Latin word *'revelation'* signifies "the act of removing the veil", which is indeed analogous to *waḥy*. Both understandings designate the act of disclosure of the hidden.

Furthermore, the realm behind the word "*waḥy*" and "revelation" has an analogous meaning in the work of Abduh, Ratzinger, and Spinoza. By defining revelation as inner-knowledge (*ʿirfān*), Abduh focuses on the cognitive and spiritual aspects of revelation, rather than on its material forms. He did not concern himself with revelation as collections of words either preserved in a heavenly tablet (*lawḥ mahfūẓ*) or in a scripture. What Abduh called "revelation as *ʿirfān* (inner-knowledge)" might resemble the idea of revelation as "more than Scripture" or the character of the New Testament as *pneuma* (spirit), not as *gramma* (letter) in Ratzinger's account. Spinoza interestingly employs a similar term, namely, "certain knowledge" or "the things that exceed the limits of natural knowledge", to describe the concept of revelation. As well as Abduh and Ratzinger, Spinoza defined the term "certain knowledge" or "the things that exceed the limits of natural knowledge[41]" not according to its material contents, but to its immateriality. Spinoza wrote, "He (God) communicates his essence to our mind without using any corporeal means".[42] However, according to Spinoza, conducting such a high level of communication necessitates a mind that is "far more outstanding and excellent than the human mind". In short, it can be inferred that Abduh, Ratzinger, Spinoza are concerned with the spiritual dimensions of revelation over its materiality.

However, in the case of Abduh, the use of *ʿirfān* might be because of the influence of Persian thinkers, i.e., through his teacher Jamaluddin Al-Afghani, who preferred to use the

word *ʿirfān* in the sense of the Gnostic tradition that refers to illuminated knowledge (*al-ʿilm al-ishrāqī/huḍūrī*) instead of acquired knowledge (*al-ʿilm al-huṣūlī*).[43] The word *ʿirfān* refers to more than the word *maʿrifah* (knowledge). In this regard, according to Mohammad Abid Al-Jābirī ([1990](#)), the term *ʿirfān* refers to knowledge that is a result of spiritual disclosure (*kashf*) and witnessing (*ayān*) rather than knowledge acquired through reason and the senses.[44] Accordingly, for Abduh, the disclosure of knowledge can be reached through direct exposure without any intermediaries or through a specific mediation.[45] In fact, if a particular intermediary mediates the revelation process, the most critical mediations are the existence of angelic beings that come to noble men, the prophets.[46]

Based on Abduh's explanation of the idea of revelation in *Risālat al-Tawḥīd*, Abduh held a sort of theological–philosophical notion inherited from the peripatetic and mystical philosophers such as Ibn Sīnā (d. 1037) and Shihābuddīn Suhrawardī (d. 1191). The influence of those philosophers appears clearly in his early work, *Risālat al-Wāridāt*. Summarizing cosmological conceptions of previous mystical philosophers, Abduh introduced a hierarchical structure of creation with different cosmic ranks (*marātib*) and degrees (*darajāt*).[47] While Abduh's predecessors such as Ibn Sina and Suhrawardi distinguished various cosmic ranks with a sophisticated angelology, Abduh introduced a simplified cosmological system. At the top of the cosmic hierarchy, there stands the Necessary Existent Being or "the truth of the truth" (*haqīqat al-haqāʾiq*), and at the bottom, the physical world (*nasūt/ ālam al-ḥayūlānī al-ṭabīʿī*).

According to Oliver Scharbrodt's account, between these two there are several cosmic ranks inhabited by angelic beings, so-called intellects and souls. First, Primal Intellect emanated from the Divine Essence and the Universal Souls (*al-nufūs al-kullīyah*), which created Particular Souls (*al-nufūs al-juzʾīyah*) who inhabit "the world of archetypes" (*al-mujarradāt*).[48] In Abduh's cosmology, as Oliver Scharbrodt ([2007](#)) elucidates, it is the Gabrielian Soul (one of the angelic beings) who is responsible for bestowing knowledge onto creation in general. In *Risālat al-Wāridāt*, Abduh stated "The bestowal of knowledge is constant process in creation from which certain individuals in the human world benefit more than others". Individuals who have achieved a certain degree of sanctity (such as saints and prophets) receive more than ordinary human beings. Gabriel, as the conveyor of divine revelations, features prominently in descriptions of Muhammad's prophetic experience. But when Muhammad describes how "he came and saw him (Gabriel) filling the whole horizon", then "this is only a metaphor (*ramz*), describing how Muhammad received knowledge from the Gabrielian Soul as part of the divine emanations".[49]

Thus, when Abduh associated the idea of revelation with knowledge, it was most likely not a kind of knowledge resulting from a discursive or speculative reasoning such as that of Islamic theologians' formulations (*mutakallimūn*), but knowledge received through the process of emanation such as that perceived by a mystic–philosopher such as Ibn Sīnā or Suhrāwardī. In turn, the justification of revelation can be obtained by means of what Abduh called "reason". However, the term reason here has a little to do with inductive and deductive reasoning; still, it is mainly associated with the mystic–philosophical reasoning by which emanative knowledge can be grasped.

In this regard, the relation between "knowledge" and "reason" in the concept of revelation in Abduh's *Risālat al-Tawḥīd* is analogous to the association of "Christ-reality" and "faith" in Ratzinger's work. It is true that Ratzinger paid much attention to the spiritual aspect of revelation (*pneuma*), but he did not discuss revelation as knowledge. Instead, he talked about revelation in terms of entering into "Christ-reality". For him, "the actual reality which occurs in Christian revelation is nothing and no other than Christ himself".[50] Following the explanation of Ratzinger, entering into "Christ-reality" can be performed through "dwelling in Christ" and "abiding presence of Christ in his Body, the Church".[51] In this context, "faith" plays an essential role in terms of accepting the idea of "Christ" as a true revelation.

Wilfred Cantwell Smith ([1993](#)) confirmed this explanation. When he searched for a parallel conception of revelation in Islam and Christianity, he did not find such a parallel

in the role of the Qur'an for Muslims and the Bible for Christians. Instead, closer to the truth of the two situations is an analogy between the role of the Qur'an in Islamic life and thought and the role of the figure of Christ in Christian life and thought. Smith mentioned "For Christian, God's central revelation is in the person of Christ, with Bible as record of that revelation. Counterpart, in the Islamic scheme of things, to the latter, the record of revelation, has been Muslim *hadīth*, the so called "Tradition", a secondary group of materials in the Islamic complex—decisive, yet secondary. Both sophisticated Muslim thinkers and comparativist Western scholars are beginning to accept this: that the genuine parallel is between the Qur'an and Christ, as two paramount motifs".[52] In short, revelation in Ratzinger's mind is much more linked with the belief in Christ-reality, whereas, for Abduh, revelation is close to the idea of knowledge of God, which can only be understood by utilizing a specific kind of reason.

As for Spinoza, revelation in the sense of things exceeding the limits of our intellect/knowledge might be revealed in various forms. He pointed out that "the things that God revealed to the Prophets were revealed to them either in words, or in visible forms, or in both words and visible forms".[53] Another possibility of revealing knowledge to the prophets is through direct communication and a true voice and images. These diverse forms, according to Spinoza, are due to the diversity of the imaginations of the prophets. He noted that "It (revelation) varied also according to the disposition of his (prophet) imagination".[54]

Interestingly, Spinoza related the acceptance of revelation to the role of imagination. According to Smith (1993), imagination is no other than a sort of reason. Imagination is a Reason (with capital "R") which can envisage a "higher" level of knowledge. Quoting nineteenth-century American philosophic theologian, C.C. Everett, Smith says" the imagination . . . is the eye of the soul ...—only (the imagination) shows us God. We receive *ḥikmah* (wisdom) through imagination, just as we receive the outward world through imagination".[55] In this regard, Spinoza's notion of revelation and its justification is closer to the idea of Abduh than that of Ratzinger. Both conceive of revelation as a certain kind of knowledge and see the importance of human intelligence in the forms of either reason or imagination. Although Spinoza also recognized the importance of the role of "faith", faith only belongs to a mass of men. As for people who have a higher quality of reason, imagination, they understand the reality of knowledge through imagination.[56]

Therefore, it might be understandable that the way of justifying the possibility of revelation in Spinoza and Abduh's work is different from that of Ratzinger. Since Abduh and Spinoza perceived the reality of revelation as a specific kind of knowledge, they believed in the efficacy of high-quality of intellect (*al-ʿuqūl al-sāmiyah*) to grasp and justify it. To refer to this high quality of intellect, Abduh used the word "reason" while Spinoza employed the term "imagination".[57] Ratzinger, conversely, conceives the reality of revelation in Christianity as Christ himself. Faith, then, plays an essential role in justifying the possibility of someone dwelling in Christ, in revelation. He pointed out that " . . . For the New Testament, faith is equivalent to the dwelling in Christ . . . the presence of Christ designated in two further ways: the individual encounters Christ and in him enters the sphere of influence of his saving power and the community of the faithful, the Church, represents Christ's continued abiding in this world in order to gather men into, and make them share, his mighty presence".[58]

## 3. How Is the Right Interpretation of Revelation Possible?

Initially, revelation is immaterial in nature. However, revelation becomes inevitably embodied in a material form in its subsequent development, since revelation should be engaged in human affairs as guidance for human beings. The material form of revelation is called "scripture". Etymologically speaking, according to Smith, scripture actually signifies what is written down. In this regard, the word "scripture" parallels its entire counterpart in Western languages. Smith makes lists of such parallelism: the cognates

*scriptura, scittura, l'Ecriture,* and *die Schrift*; in the preceding Greek *he graphe, hai graphai*; and the Hebrew *ketuvim*.

Similarly, in all its forms, the word "Bible", the Greek *biblia*, and the Hebrew *sepher* and *ketab* signify the book (the written words).[59] Human involvement is inevitable in the process of the embodiment of revelation, changing from an immaterial entity into a material form. It is humans who transform revelation into a scripture, from immateriality into a material text. It is also humans who conceive of scripture not as a worldly text, but as a sacred text. Scripture becomes sacred because humans perceive it as a sacred; thus, they treat it accordingly. In this regard, I agree with Wilfred Cantwell Smith's statement that "People—a given community—make a text into scripture, or keep it scripture; by treating it in certain way ... scripture is human activity".[60]

After revelation becomes a written text, one might ask whether its status should still be considered a revelation that conveys a divine truth of knowledge. The answer to these questions varies. It depends on the understanding and interpretation of the scripture. Some modern scholars might view a scripture as a mere text which could be approached through literary theories such as structuralism, deconstruction, hermeneutics, and discourse analysis. They assume that objectivity is the best way to understand phenomena, even considering it the only correct way.[61] Scripture, in turn, is detached from its status as a sacred text or an embodiment of revelation. For them, the scripture is no longer revelation. Nevertheless, according to some scholars of religion, a specific treatment and approach towards religious texts is necessary. W.C. Smith, for example, considers a scripture not to be a text.[62] I concur with Smith in the sense that, at least, scripture is not an ordinary text. Although there is human involvement in the process of the formation of the scripture, its status is more than that of a regular text. There are elements of sacredness in scripture; therefore, it cannot be treated as an ordinary book. First, it is sacred in terms of its divine source (God), and, second, scripture is sacred because people perceive it to be so.

Theologians including Abduh, Ratzinger, and Spinoza are in agreement in their view of scripture as a specific text which has elements of sacredness. They also believe in a sort of efficacy of scripture in delivering divine knowledge in different degrees. Accordingly, interpretation of the scripture is very important because it not only aims to gain an objective meaning of the text, but, more importantly, it is also directed towards achieving the highest level of knowledge. However, is it possible to attain the realm of revelation through reading scripture? Ratzinger, Spinoza, and Abduh agreed that humans are endowed with a certain ability to experience the realm of revelation through a right interpretation of scripture. However, they have different opinions regarding how humans can come to the right interpretation.

Ratzinger maintained that the explication of Christ-reality is that revelation occurs in the proclamation of the gospel. Therefore, if one wants to understand and enter to Christ-reality, one should carry out at least one of two kinds of interpretations. First, one interprets the Old Testament of the Christ-event as oriented towards that event. Second, there is the interpretation of the Christ-event itself based on *pneuma*, which means it is based on the Church's presence.[63] Since Ratzinger believes that Christ is not dead but living, he favored the latter form of interpretation over the former. For him, Christ is living and present in His Church, which is His Body and in which His Spirit is active.[64] Therefore, the interpretation of the Christ-event on the basis of Church's presence requires faith in the living Christ, by which humans can have an authoritative interpretation of revelation. This context of interpretation places *fide* above *scriptura*, that is to say, "the rule of faith above the details of what is written".[65] In matters of faith, Ratzinger elucidated that there is a double criterion that must be affirmed. He stated, "On the one hand, there is what in the ancient Church was called "the rule of faith" and with it the regulative function of the official witness as against scripture and its interpretation, that *praescriptio* of the rightful owner of scripture, and this excludes any willful playing of the scripture against the Church. On the other hand, there is also the limit set by *littera scipturae,* the historically ascertainable literal meaning of scripture which certainly represents no absolute criterion subsisting in and for

itself within the counter point of faith and knowledge, but does nevertheless represent a relatively independent criterion".[66] In this regard, Ratzinger tended to believe that the authoritative interpretation that can lead to the realm of revelation (Christ-reality) is the Church's interpretation of the Christ-event to the scripture rather than the interpretations of individual theologians.[67]

While Ratzinger believed in the rule of faith and the Church's power in the activity of interpretation, Spinoza seemed to put more trust in the efficacy of scripture itself with the aim of gaining true knowledge. Spinoza criticized theologians who have mainly been anxious to twist their own inventions and beliefs out of the Sacred Text and fortify them with divine authority.[68] They used their speculative reasoning recklessly to interpret the Sacred Text and to read the mind of the Holy Spirit. Nevertheless, for Spinoza, all knowledge of scripture cannot be sought through something outside scripture such as reason, intellect, or miracle. Knowledge must be sought only from scripture itself because the only true method of interpreting scripture is through scripture itself.[69] Spinoza gives an example of the divinity of God and of scripture. He noted that the divinity of God cannot be proven by miracles. Thus, the divinity of scripture must be established only by the fact that it teaches true virtue. However, such divinity "can only be established only by Scripture".[70]

According to Spinoza, the basic notion of the interpretation of scripture does not differ from that of the method of interpreting nature. If the study of nature consists of putting together a history of nature from which one can obtain specific data and infer the natural definitions of natural things, interpreting scripture is similar. Spinoza pointed out that knowing the history of scripture is necessary "to infer the mind of the authors of Scripture (with the capital "S")". By having historical accounts of the Scripture, one will interpret the Scripture without "any danger of error" and will be able to discuss "the things which surpass our grasp as safely those we know by natural light".[71]

However, it is rather problematic when someone encounters seemingly ambiguous and contradictory contents in scripture. Although theologians might offer certain interpretations based on "reason", Spinoza would deny this and still believes in the sufficiency of scripture. For example, Moses stated that *God is a fire* and *God is jealous.* He preferred not to read this immediately and interpreted it metaphorically. He avoided going too far from the literal meaning of the scripture. First, he would ask whether this sentence admits another meaning beyond the literal one, whether the term *fire* signifies something other than natural life. If not found, then those sayings of Moses would be irreconcilable; thus, one should suspend judgment about them. However, because the term *fire* is also taken for anger and jealousy, these sentences of Moses are easily reconciled. Finally, one could legitimately argue that these two sentences, *God is a fire* and *God is jealous,* are one and the same sentence.[72] In other words, Spinoza's approach to dealing with such ambiguity and contradiction is by placing all the ambiguous teachings of scripture into their context. Spinoza pointed out that "Whatever is found to be obscure or ambiguous in the texts . . . must be explained and determined according to the universal teaching of the scripture. But if we find any contradictions, we must see on what occasion, and at what time, and for whom they are written".[73]

From the above explanations, it can be said that Spinoza was reluctant to use reason to interpret scripture. He warned people against using reason because it brings about certain biases. He said that "We must take a great care, so long as we are looking for the meaning of scripture, not to be preoccupied with our own reasoning, insofar as it is founded in principles of natural knowledge (not to mention our prejudices)".[74] Instead of employing reason (rational interpretation without referring to the context of scripture), Spinoza relies heavily on the sufficiency of scripture. Scripture and all its properties (history, language, and structures) are considered efficacious in the disclosure of the reality of the knowledge behind the written words. Spinoza believed that scripture itself can lead humans to experience the realm of revelation.

Apart from Ratzinger and Spinoza, Abduh relied more on the ability of reason to interpret the scripture and, at the same time, to allow humans to experience the realm of revelation. He saw that reason (*aql*) and scripture (*naql*) are the twin bindings that give Islam unity.[75] In *Risālat al-Tawḥīd,* he wrote that "One thing that we have to believe is that the religion of Islam is religion of *tawḥīd* (oneness of God) . . . reason is its strongest support and scripture is its most important element".[76] According to Livingston (1995), Abduh held the idea that reason and scripture, although they exist in their own self-contained spheres of truth, find mutual support and reaffirmation.[77] This interpretation is valid since Abduh himself maintained that whatever contradiction might exist between reason and scripture, they are superficial and not worthy of concern.[78]

However, when there is an apparent contradiction between them, he was inclined to reason to undertake a rational and acceptable interpretation of the scripture. He wrote in *Al-Islām wa al-Naṣrānīyah*, "If there is a contradiction between reason (*aql*) and scripture (*naql*), it is better to take what is proven by reason. Then, two remaining possible attitudes towards scripture exist. First, accepting the validity of scripture with the awareness of a weak comprehension of it, then submitting all of things to the realm of knowledge of God. Second, interpreting the scripture by maintaining the principles of language until the meaning of the scripture corresponds to the things established by reason".[79]

In the assessment of Zaki Badawi (1978), moreover, Abduh considered the message of prophets to be complementary to reason. For him, the message cannot possibly contradict reason and cannot supersede it. He asked "How could the place of reason be denied when the proofs of revelation must be sifted and evaluated by the reason?" But, Abduh continued, "Once reason concludes that the claimant to the prophecy is truthful, reason must accept all information given by him. It must do so even if the nature of some of them is beyond it. In other words, revelation can be accepted if it is above reason, but not if it is contradiction with reason. Suppose it appears to contradict itself or reason, we must not accept this apparent contradiction when we have the choice either of interpreting revelation so as to arrive at consistent meaning or else to spare our selves the effort and simply rely on Allah".[80]

In brief, the three religious scholars (Ratzinger, Spinoza, and Abduh) agreed that religious scriptures have specific knowledge that can bring people to the realm of revelation, beyond the boundaries of the written word. However, they disagreed in determining the best way to approach and interpret scripture. Ratzinger was confident with the authority of the Church in conducting such interpretations because the Church represents the body of the Christ. Spinoza underlines the efficacy and sufficiency of scripture itself in terms of gaining true knowledge from the scripture. By doing so, one can find a way to know the nature of revelation. Muhammad Abduh believed in the power of reason. It is reason that becomes the most substantial support of religion and, in turn, can interpret scripture to achieve the true meaning of revelation.

## 4. Conclusions

Despite the different backgrounds of Abduh, Ratzinger, and Spinoza (the first was from the reformist group in Islam, the second is from the Roman Catholic establishment, the last was a Jewish philosopher), their thoughts on revelation are identical in terms of perceiving revelation in the sense of a spiritual dimension. It might be true that Muhammad Abduh frequently criticized Christianity for its adherence towards the Church and disregarding the role of reason in faith.[81] However, after considering the notion of revelation proposed by Ratzinger, it appears that Abduh's idea of revelation is homologous to the *pneuma* (spirit) of the Christian scripture and phenomena exceeding the limits of knowledge in Spinoza's explanation.

In Ratzinger's account, if someone wants to grasp the nature of revelation fully, he should dwell on the reality of *pneuma*, which is Christ-reality, or, for the Church, as the enduring and authoritative presence of Christ. The most critical element here is faith. Similarly, when Spinoza described the nature of revelation as certain knowledge that

requires more than an ordinary mind to perceive it, he turned to so-called "imagination" and "a sheer faith" to justify the possibility of revelation.

Analogously, to understand the nature of revelation, for Abduh, one should immerse oneself in the reality of illuminated inner-knowledge through intellectual and spiritual exercises. Reason (*'aql*), then, becomes highly significant. Indeed, on occasions when there is a contradiction between reason (*'aql*) and revelation (*naql*), Abduh would prefer to use reason over revelation. He would interpret revelation until the meaning became suitable with the use of reason.[82] However, reason (*'aql*) in this regard is not only used in terms of acquiring demonstrative knowledge (*al-burhānī*) but also illuminative knowledge (*maḥḍ al-fayḍ al-ilāhī*).[83] Reason that is harnessed to gain such illuminative knowledge, spiritual disclosure (*kashf*), and witnessing (*ayān*) is actually closer to the reality of faith than reason used in deductive reasoning.

The situation becomes somewhat problematic when revelation is embodied in the form of the written word, in scripture. Is it possible for someone to know the reality of revelation through an understanding of scripture? If it is possible, how can they achieve this? Ratzinger, Spinoza, and Abduh gave different accounts regarding this matter. Although they tended to share opinions on the possibility of human beings to know the reality of revelation, they had different approaches to achieving that goal. Ratzinger gave trust and authority to the Church in making scriptural interpretation; Spinoza believed in the self-explaining ability of scripture; while Abduh saw the superiority of reason over scripture, although he claimed that there was no contradiction between them.

However, this comparison gives a fresh understanding of Muhammad Abduh's thought written in *Risālat al-Tawḥīd*, especially regarding the notion of revelation. First is the idea that revelation is immaterial in nature. Ratzinger and Spinoza also shared this idea, although their expressions of this were different. Second, Abduh insisted that the possibility of revelation could be justified by reason. The notion of reason in this regard is closer to reason as a product of emanation (Gnostic) than reason used in inductive or deductive reasoning. Third, the authority of scriptural interpretation lies neither in religious institutions nor in religious scholars, but in reason—guided reason. Scripture cannot speak without the interpretation of reason.

Abduh's other counterparts shared his notion of revelation as an immaterial entity in a larger context. Ratzinger called it the *pneuma* (spirit) that resides behind the letter (*gramma*). Spinoza named it as "the things that exceed the limits of our intellect or knowledge". Analogously, if Abduh employed reason as a justification with regard to revelation, Ratzinger chose "the rule of faith" and Spinoza preferred to use "a sheer faith" of revelation. After gaining an adequate understanding of the notion of revelation among these religious traditions, mutual respect and constructive dialogue could be enhanced to build a tolerant and peaceful world.

**Author Contributions:** Writing—original draft, M.S.A.W.; Writing—review & editing, A.R.B.Z. All authors have read and agreed to the published version of the manuscript.

**Funding:** This research received funding from LP3M (Research and Innovation Board), Universitas Muhammadiyah Yogyakarta.

**Conflicts of Interest:** The authors declare no conflict of interest.

## Notes

[1]   (Kung 2007).

[2]   Kung, *Islam: Past, Present and Future*.

[3]   (Ameri 2013).

[4]   Muhammad Abduh is Egyptian Islamic Reformer (1849–1905). The context for Muhamad Abduh's thought was the inner decay and the need of inner revival in Islamic society. At the time, there was a gap between what Islamic society should be and what it had become. To overcome this problem, an early Islamic reformist Tahtawi, called to adopt European sciences. Instead of merely adopting European sciences like Tahtawi, Muhammad Abduh in fact endeavored to seek a religious ground for reviving the

glory of Islam. By arguing that Islam is compatible with modernity and revelation is not opposed to reason, Abduh sought to demonstrate how Muslims as individuals and society should behave in dealing with modernity.

5　　(Shabir and Susilo 2018, p. 6).

6　　(Fogg 2015, p. 2).

7　　(Scharbrodt 2007, p. 1).

8　　(Aydin 2005, p. 24).

9　　(Livingston 1995, pp. 3–4).

10　　(Seferta 1985, p. 2).

11　　(Ratzinger 2006). His papacy began on 19 April 2005. I am interested to compare Abduh's view on revelation with Ratzinger, who is also prolific scholar and Catholic theologian, because Ratzinger's clear position on the notion of revelation and his engagement with Muslim world in this speech at the University of Regensburg, in 2006.

12　　In Theodore Khury's work, Muslim's faith is spread by sword and Muslim's God is transcendental, not bound up with rational category, and even not bound up by His own word. See (Ratzinger 2006, pp. 2–4).

13　　(Paul 1998).

14　　(Leezenberg 2021, p. 2; Widigdo 2020b, pp. 123–46).

15　　(Mirzaei 2021, p. 2).

16　　(Husserl 1999; Kockelmans 1994; Butler 2016, p. 11).

17　　(Geldhof 2007, p. 2).

18　　(Ratzinger 1966, p. 35).

19　　(De Spinoza 1994, p. 10).

20　　Spinoza, *A Spinoza Reader: The Ethics and Other Works*, 10–11.

21　　Spinoza, *A Spinoza Reader: The Ethics and Other Works*, 10.

22　　(Abduh 1994, p. 102).

23　　Revelation in this sense is recognized as a source of knowledge in Islamic epistemology. (Atmaja and Mustopa 2020, pp. 30–32).

24　　(Knysh 1992, p. 632).

25　　Abduh once said about his teacher, "al-Afghani was personally devout, with a tendency towards mysticism in his thought . . . ". (Hourani 1983).

26　　(Hourani 1983, pp. 131–32).

27　　Ratzinger, *Revelation and Tradition*, 40.

28　　Ratzinger, *Revelation and Tradition*, 36.

29　　Ratzinger, *Revelation and Tradition*, 40.

30　　Spinoza, *A Spinoza Reader: The Ethics and Other Works*, 15.

31　　Spinoza, *A Spinoza Reader: The Ethics and Other Works*, 16.

32　　Spinoza, *A Spinoza Reader: The Ethics and Other Works*, 17.

33　　Spinoza, *A Spinoza Reader: The Ethics and Other Works*, 10.

34　　Spinoza, *A Spinoza Reader: The Ethics and Other Works*, 42.

35　　Abduh, *Risālat al-Tawḥīd*, 104.

36　　Abduh, *Risālat al-Tawḥīd*, 105.

37　　(Abduh 1988, p. 69).

38　　Abduh, *Risālat al-Tawḥīd*, 102.

39　　Abduh, *Risālat al-Tawḥīd*, 101.

40　　Abduh, *Risālat al-Tawḥīd*, 101.

41　　Spinoza, *A Spinoza Reader: The Ethics and Other Works*, 11.

42　　Spinoza, *A Spinoza Reader: The Ethics and Other Works*, 14.

43　　See the discussion of these two forms of knowledge, knowledge by presence (*al-ʿilm al-huḍūrī*) and knowledge by correspondence (*al-ʿilm al-huṣūlī*), in (**?**; Fattah Santoso and Khoirudin 2018, pp. 81–84; Widigdo 2020a).

44　　(Al-Jābirī 1990, p. 251).

45　　Abduh, *Risālat al-Tawḥīd*, 102.

46　　Abduh, *Risālat al-Tawḥīd*, 105.

47　　Scharbrodt, "The Salafiyya and Sufism: Muhammad Abduh and his Risālat al-Wāridāt (Treatise on Mystical Inspirations)", 101; (Abduh 1925, p. 14).

48  Scharbrodt, "The Salafiyya and Sufism: Muhammad Abduh and his Risālat al-Wāridāt (Treatise on Mystical Inspirations)".

49  Scharbrodt, "The Salafiyya and Sufism: Muhammad Abduh and his Risālat al-Wāridāt (Treatise on Mystical Inspirations)", 101; Abduh, *Risālat al-Wāridāt fī naẓarīyāt al-mutakallimīn wa al-ṣūfīyah*, 15.

50  Ratzinger, *Revelation and Tradition*, 40.

51  Ratzinger, *Revelation and Tradition*, 46.

52  (Smith 1993, p. 46).

53  Spinoza, *A Spinoza Reader: The Ethics and Other Works*, 11.

54  Spinoza, *A Spinoza Reader: The Ethics and Other Works*, 18.

55  Smith, *What is Scripture?*, cf. 29, 361.

56  This idea of imaginative faculty that can receive divine knowledge or revelation is previously held by Jewish philosopher, Maimonides (d. 1204), who was influenced by Muslim peripatetic philosopher, Abū Naṣr al-Fārābī (d. 950). See the discussion on Maimonides' idea of revelation and prophecy and his peripatetic predecessors in Widigdo, "Philosophical and Religious Justification of Prophecy: A Comparative Analysis Between al-Ghazālī and Maimonides' Accounts of Prophecy".

57  The concept of "reason" in Abduh (d. 1905) and "imagination" in Spinoza (d. 1677) was also addressed by an influential philosopher who lived in between the period of Abduh and Spinoza's career, namely Immanuel Kant (d. 1804). Kant addressed the relationship between "reason" and "imagination" primarily in the first critique, *Critique of Pure Reason* (Kant 2007), and the third critique, *Critique of Judgment* (Kant 2000). (Ayas Onol 2015) summarizes Kant's explanation regarding how reason and imagination comprehend sublime experiences as follows. Imagination as a faculty of the soul appears only in relation to the faculty of understanding. To comprehend sublime objects and experiences, the idea of absolutely great or absolutely strong, the imagination needs "reason". The "reason" as the highest faculty in human mind accompanies the imagination in the realm of sublime and makes the imagination expand further. Ayas concludes Kant's thought", Thus, by this extension of the imagination (*Erweiterung*), the comprehension of the sublime object is achieved". (Ayas Onol 2015, p. 63). In this regard, Spinoza's "imagination" and Abduh's "reason" can be understood as faculties of human mind in Kantian sense that are essential to comprehend the sublime reality of revelation as *a priori* knowledge.

58  Ratzinger, *Revelation and Tradition*, 40.

59  Smith, *What is Scripture?*, 7.

60  Smith, *What is Scripture?*, 18.

61  Smith, *What is Scripture?*, 222.

62  Smith, *What is Scripture?*, 223.

63  Ratzinger, *Revelation and Tradition*, 41–42.

64  Ratzinger, *Revelation and Tradition*, 42.

65  Ratzinger, *Revelation and Tradition*, 45.

66  Ratzinger, *Revelation and Tradition*, 48–49.

67  Ratzinger, *Revelation and Tradition*, 44–45.

68  Spinoza, *A Spinoza Reader: The Ethics and Other Works*, 40.

69  Spinoza, *A Spinoza Reader: The Ethics and Other Works*, 46–47.

70  Spinoza, *A Spinoza Reader: The Ethics and Other Works*, 42. The scripture with capital "S" in Spinoza's thought here is used to attach an element of sanctity of the scripture as an embodiment of revelation.

71  Spinoza, *A Spinoza Reader: The Ethics and Other Works*, 41.

72  Spinoza, *A Spinoza Reader: The Ethics and Other Works*, 44.

73  Spinoza, *A Spinoza Reader: The Ethics and Other Works*, 45.

74  Spinoza, *A Spinoza Reader: The Ethics and Other Works*, 43.

75  Livingston, "Muhammad Abduh on Science", 225.

76  Abduh, *Risālat al-Tawḥīd*, 23.

77  (Rachman 2013, p. 9)

78  Livingston, "Muhammad Abduh on Science".

79  Abduh, *Al-Islām wa al-naṣrānīyah maʿa al-ʿilm wa al-madanīyah*, 70.

80  (Badawi 1978, pp. 58–59). A further discussion of different forms of "reason" and reasoning are discussed in Islamic intellectual tradition, (Muslih 2018, p. 14).

81  Abduh, *Al-Islām wa al-naṣrānīyah maʿa al-ʿilm wa al-madanīyah*, 35.

82  Abduh, *Al-Islām wa al-naṣrānīyah maʿa al-ʿilm wa al-madanīyah*, 70.

83  Abduh, *Risālat al-Tawḥīd*, 104.

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
