# Peer review of "Revelation in the Muslim, Christian, and Jewish Traditions: A Critical Analysis of Muhammad Abduh’s Notion of Revelation through the Lens of Joseph Ratzinger and Benedict De Spinoza"

_religions, doi:10.3390/rel12090718_

Round 1

Reviewer 1 Report

This is an extremely engaging and promising essay that beautifully balances assessment of Catholic, Jewish, and Muslim thought on the topic of revelation. It is well-organised and eminently reasonable. 

There is a major omission, however, which absolutely needs to be addressed, if this piece is to be published. This omission is the lack of engagement with John Paul II's encyclical Fides et Ratio (1998). There are several reasons to engage this more recently published text, rather than Joseph Ratzinger's Revelation and Tradition (1966), as the example of Catholic thought within this specific argument. Primarily, there are much closer, analogous, parallels between Fides et Ratio and the work of Muhammad Abduh. This is because in his encyclical John Paul II sought to illustrate how faith and reason can find what the proposed essay terms "mutual support and reaffirmation" ("Revelation in Muslim, Christian, and Jewish Tradition", 10). Secondly, this proposed essay promises to examine the question of reason and revelation through "a phenomenological lens", and John Paul II was academically trained in phenomenology. Many scholars see him as primarily a phenomenologist. Thirdly, in Fides et Ratio, John Paul II affirms the knowledge provided by what he terms the "natural sciences" at multiple points, which creates a solid parallel with Spinoza's affirmation of what the proposed essay terms "data" from which scientists "infer the natural definitions of natural things" (10). If the goal of this essay is to cultivate and nurture inter-faith dialogue, Fides et Ratio is the most logical choice for finding even greater common ground with both Abduh and Spinoza. So, it would be best to cut the references to Revelation and Tradition altogether and replace them with references to Fides et Ratio (Faith and Reason). 

In terms of a more minor revision, a slight structural adjustment is also needed. The Gnostic sources of Abduh's ideas become apparent to the reader well before Gnosticism appears in the argument. So, I would also suggest mentioning the influence of Gnosticism on Abduh earlier on, within the first few pages of the essay.

Additionally, the thought of philosopher Immanuel Kant warrants mentioning in at least a footnote. In terms of intellectual history, Kant's ideas about reason and imagination emerged in-between the time of Spinoza and Abduh, so they could provide an important bridge historically.  

Author Response

Dear Reviewer I,

Thank you for your valuable review and suggestions to improve the article's argument. Please find my response in the attachment

Best,

Author

Reviewer 2 Report

This text presents an interesting analysis of three notable theologians and their arguments in relation to revelation and use of human reason. It can be an important contribution to the field.

The text can however be improved by considering four different ways of making the text into a publishable article in Religions:

  1. The text requires better streamlining of the analysis, which seems to run through the entire length of the article, not only towards the end of the text. This is not conventional, and it is exciting to read, however, a better structure can make it a more compelling read. For instance, gathering the three arguments of definition of revelation, definition/understanding of human reason, and interpretation(all under different headings could be a way to go).
  2. A more extensive intellectual/theological context could be included as to make better sense of these three theologians' thinking (they are after all, writing in very different historical periods)
  3. There is little focus on the issue of religious tolerance and inter-religious communication, and that is stated in the abstract and slightly in the beginning of the text, and later very briefly in the conclusion.
  4. Revise the language and make it more precise - the concept of revelation and some other words are used in different ways in different parts of the text. For instance at one point scripture is revelation and in another part revelation is more than that etc. Consistency is extremely important in a text this dense.

Author Response

Dear Reviewer II,

Thank you for your valuable review and suggestions to improve the article's argument. Please find my response in the attachment

Best,

Author

Round 2

Reviewer 1 Report

This article is much improved. Adding references of Pope St. John Paul's encyclical on Faith and Reason, as well as Immanuel Kant's work on reason, judgment, imagination and the sublime provides a broader context.

There are some English language issues remaining (missing words, grammatical glitches, etc.). So, the article needs to be thoroughly copy-edited before it is published.